# Challenges in AAV-Based Retinal Gene Therapies and the Role of Magnetic Nanoparticle Platforms

**DOI:** 10.3390/jcm13237385

**Published:** 2024-12-04

**Authors:** Oliver Siontas, Seungkuk Ahn

**Affiliations:** 1Eidgenössische Technische Hochschule (ETH) Zürich, Department of Biosystems Science and Engineering, 4056 Basel, Switzerland; 2UCD Charles Institute of Dermatology, School of Medicine, University College Dublin, D04 V1W8 Dublin, Ireland

**Keywords:** eye diseases, retinal gene therapy, adeno-associated virus, magnetic nanoparticles

## Abstract

Retinal diseases, leading to various visual impairments and blindness, are on the rise. However, the advancement of retinal gene therapies offers new hope for treatment of such diseases. Among different vector systems for conferring therapeutic genetic load to retinal cells, adeno-associated viruses (AAVs) have been most intensively explored and have already successfully gained multiple clinical approvals. AAV-based retinal gene therapies have shown great promise in treating retinal disorders, but usually rely on the heavily disruptive administration methods such as subretinal injection. This is because the clinically well-established, minimally invasive alternative of intravitreal injection (IVI) necessitates AAVs to traverse the retinal inner limiting membrane (ILM), which is hard to penetrate in higher eye models, like human or porcine eyes. Additionally, AAVs’ natural transduction preference, known as tropism, is commonly not specific to cells of only one target retinal layer, which is another ongoing challenge in retinal gene therapy. This review examines strategies to overcome these obstacles with a focus on the potential of magnetic nanoparticles (MNPs) for improved retinal AAV delivery.

## 1. Introduction

Globally, retinal diseases play a significant role in causing ocular morbidity and visual impairment. Studies conducted on populations have demonstrated a prevalence of retinal disorders ranging from 5.35% to 21.02% in individuals aged 40 years and above [1]. In developed nations, retinal diseases are the leading cause of irreversible blindness [2]. Additionally, the two most prevalent blindness-inducing retinal disorders, neovascular age-related macular degeneration (nAMD) and diabetic retinopathy, are projected to increase in total case numbers by about one to two hundred million in the next twenty years [3,4]. Nevertheless, recent outcomes in the field of retinal gene therapies aiming to confer cure to non-hereditary and inherited retinal disorders (IRDs) stir up hopes for a better prospect into the future. This development must be considered a result of a global medical trend in which the novel advancement of gene therapy specifically excels in treating inherited and untreatable diseases since it allows the introduction of therapeutic genetic material into disease cells to correct underlying previously inaccessible genomic defects [5].

In late 2017, the U.S. Food and Drug Administration (FDA) approved Luxturna (voretigene neparvovec, Spark Therapeutics, USA) the first adeno-associated virus (AAV)-based gene-therapeutic product for the treatment of a recessive IRD called Leber congenital amaurosis-2 (LCA2) [5]. Notably, the treatment was shown to significantly increase vision and to meet safety requirements [6,7]. This milestone approval marked the dawn of gene therapy for retinal conditions and ophthalmology. The Luxturna treatment functions by gene augmentation—the strategy of replacing a disease gene, in this case RPE65, with a healthy copy to restore function—which is to date the most extensively studied gene therapy approach for retinal disease treatments [8]. However, other methodologies are also being explored in clinical studies, such as inducing the expression of therapeutic proteins e.g., for non-hereditary nAMD (NCT01024998), suppressing disease gene expression e.g., for the IRD Retinitis Pigmentosa (NCT04123626), in vivo precision editing of disease-causing mutations and interfering splicing processes e.g., for the IRD LCA10 (NCT03872479 and NCT03913143).

Although all the above highlighted examples differ greatly in their genetic strategies, they unanimously rely on the initial mobilization and transfer of genetic material into the target disease cells. Generally, the delivery of genetic material into cells can be achieved through the utilization of vectors that are either of viral or non-viral origin. Noteworthy examples for the latter would be lipid-based carriers like liposomes, composed of cholesterol and phospholipids that can merge with lipidic cellular membranes to release their genomic content into the cell [9]. Additionally, biopolymers such as polylactic acid, or polyethylenimine, are being investigated as potential vectors for genetic therapy [9]. Although non-viral techniques obtain low immunogenicity and can store large genetic cargo, they are incapable to replicate which implies the necessity for repetitive administrations and are less efficient in targeting cells in vivo compared to viral methods [10]. Therefore, as well as due to recent progresses in virus engineering and serotype selection for enhanced tissue specificity—particularly in the field of AAV research [11,12,13]—viruses became the most used delivery platforms for gene therapies in the last years [14].

This review underlines the key challenges of utilizing viral vectors for retinal gene therapy, with a particular emphasis on AAV-based approaches. We also explore the potential of magnetic nanoparticles (MNPs) as remotely controllable AAV delivery platforms to overcome such obstacles.

## 2. Viral Vectors in Retinal Gene Therapy

In the context of viral gene therapy, the eye particularly is an attractive target due its immune-privileged nature. In contrast to other parts of the body, immune responses to viruses or other foreign substances are limited in the eye, which supports its compatibility with gene therapies [8,15]. Additionally, the eye bears advantageous anatomical properties, such as that the presence of a blood–retinal barrier that prevents vector leakage into the bloodstream. Additionally, due to its relatively enclosed structure and small size the eye needs only low vector dosages for efficient delivery [8]. Furthermore, the retina is ideal for gene therapy because it is easily visualized, lacks lymphatic vessels and a direct blood network in the outer layers, and its cells do not divide after birth, preserving transgene expression when edited [16].

Viruses are non-living particles that carry genetic material (DNA or RNA) in protein envelopes, called capsids, which in some cases are further engulfed by an outside shell of lipids. They possess a natural ability to infect human cells and integrate their genetic material into the nucleus of the host cell, making them highly efficient vehicles for genetic modification [17]. While multiple viruses can be loaded with therapeutic genomic material, their type of compatible cargo, mode of transduction and host cell specificity may differ substantially [18]. The latter is also called tropism and describes the preferences of viruses to transduce cell types selectively, which are typically determined by the composition of their capsid proteins [19]. However, tropism diversity among viral variants broadens the landscape of possible therapeutic applications for viral vectors since it allows one to address different cell types or tissues only via the choice of virus. For instance, various neural cells in the multilayered retinal tissue—composed of the inner limiting membrane (ILM), ganglion cell layer (GCL), inner plexiform and nuclear layers (IPL and INL), outer retina layers containing the outer nuclear layer (ONL), photoreceptor layer (PL), and retinal pigment epithelium (RPE)—can be individually targeted only by the choice of respective viral vectors that obtain matching tropism specificities (Figure 1).

Lentiviruses contain single-stranded RNA and have the capability to integrate their genetic material directly into the chromosomes of host cells. This integration process allows lentiviruses to replicate continuously, even in dividing cells, following a single administration. Therefore, they can sustain stable long-term expression of the transgene after efficient initial transduction. Moreover, lentiviruses are well-suited to stably deliver genes of up to 10 kilobases (kb) in size. However, it is important to note that the use of lentiviruses for gene delivery carries the risk of insertional mutagenesis, which can lead to the disruption of specific genes and potentially impairs cell viability or contributes to tumor development [20].

In contrast, adenoviruses bear double-stranded DNA and are known for their ability to accommodate a larger amount of genetic material compared to other viral vectors, with a loading capacity of up to 37 kb. Adenoviral vectors are capable of infecting both dividing and non-dividing cells. Different to lentiviruses, the genetic material of adenoviruses remains separate from the host cell genome in extrachromosomal episomes after transduction, which minimizes the risk of insertional mutagenesis [18]. Adenoviruses, especially serotypes 2 and 5, have been extensively utilized in gene therapy research and were shown to effectively transduce RPE cells and, in some cases, photoreceptor cells in retinal tissue [21,22,23,24]. However, it should be noted that adenoviral gene delivery does not integrate the genetic material into the target cell genome, leading to its dilution with each cell division in dividing cells. Additionally, the immune response elicited by adenoviruses in the host organism can pose challenges, as the immune system may eliminate host cells expressing adenovirus proteins through cytotoxic T cell intervention, thereby removing the genetic information necessary for successful transduction [18]. To circumvent these limitations, helper-dependent adenoviruses capable of escaping the host’s immune response have been developed [25,26].

In the field of retinal and ocular gene therapy, adeno-associated viral vectors (AAVs) have specifically gained significant prominence. By 2018, about 70% of gene therapy clinical trials for ocular diseases were conducted using AAVs, and serotype 2 was the leading vector [5]. For retinal diseases particularly, more than 50 AAV-mediated clinical trials have been instigated to this date (clinicaltrials.gov; search term “(AAV OR adeno associated virus OR adeno associated vector) AND (retina OR retinal)”).

AAVs belong to the Parvoviridae family and are 25 nm non-enveloped viruses containing single-stranded DNA (ssDNA) [27,28]. Unlike other viruses, AAVs require a helper virus for replication which renders them non-pathogenic. They exhibit the capability to infect both dividing and non-dividing cells and, like adenoviruses, their genetic material is deposited in episomes after transduction. With a genome size of approximately 4.8 kb, the packaging capacity of AAVs is comparably limited [29]. However, AAVs have gained a leading role in retinal gene therapy, likely due to their relatively small size and resulting ability to spread through retinal tissue when directly injected into it [28]. Additionally, AAV’s low immunogenicity allows for repetitive administrations and long-term expression of transgenes [30]. Lastly, a multitude of different AAV serotypes have been identified with varying cellular tropism [19,31,32]. The serotypes also differ in their transfection efficacy and kinetics. AAV2 was the first serotype utilized for retinal gene transfer. Whilst AAV2 efficiently transduces Ganglion cells, for instance, it only poorly transduces photoreceptors [33,34,35]. Fortunately, each AAV has its own external capsid protein shell, which can be recombinantly exchanged between different serotypes [36,37]. Therefore, different hybrid recombinant AAVs (rAAVs), so-called pseudo-types, comprising a capsid of one type and the genome of another, were designed to engineer immune responses against the capsids epitopes and further specify tropism to individual retinal cell types [27,36]. Pseudo-types generated by this capsid exchange are named e.g., AAV2/1, where the first number refers to the genomic and the second to the capsid origin. The nine natural serotypes AAV1 to 9 are most used to produce pseudo-type rAAVs (Table 1). Moreover, rAAVs that are self-complementary versions of conventional AAVs, so-called scAAVs, are often used since they are capable of inducing double-stranded DNA (dsDNA) association autonomously, which significantly enhances robust transgene expression [38,39].

Thanks to the technique of pseudo-typing, variants were generated that, for instance, more preferably transduce cells of the PL and RPE. Most vision-disturbing retinal disorders arise from malfunction occurring in cells of the PL and RPE, where the light-detecting rod and cone photoreceptor cells reside on a support of layered pigmented cells [35]. Thus, in an effort to heal retinal diseases, these layers are prime targets for gene therapies [40]. Virtually every pseudo-type of AAV2 is able to minimally transduce all retinal layers when injected directly into the tissue [28,31,41,42,43]. However, some pseudo-types, namely AAV2/1, AAV2/4, and AAV2/6 or AAV2/5, 2/7, 2/8, and 2/9 have been demonstrated to transduce RPE cells or photoreceptor cells, respectively, with significantly higher efficiency (Table 1). Conversely, others, like AAV2/2, show higher tropism to cells of the inner layers, such as Müller or Ganglion cells.

An additional technique for limiting AAV transduction to a subset of retinal cells is promoter engineering. For instance, several AAV promoter sequences have been investigated that drive expression in different photoreceptor cell types. Cone-specific transgene expression can be achieved with promoters like cone arrestin, blue opsin, or red/green opsin [44,45,46,47]. Rod-specific expression relies on rhodopsin promoters from different species [48,49]. Other promoters, such as rhodopsin kinase 1 and interphotoreceptor retinoid-binding protein, drive expression in both photoreceptor types [50,51]. Next to the PL, cell types of other layers have also been addressed with promoter variants, such as RPE cells (RPE65, VMD2) and bipolar cells (GRM6/SV40) [52,53,54,55]. Additionally, ubiquitous promoters, like CMV or CAG, are widely used since they induce high expression levels [43,56,57,58,59,60,61,62]. In combination with a respective pseudo-type, they can be exploited to drive intensified expression, specifically in a target population of retinal cells.

Finally, multiple studies have explored approaches for engineering capsid proteins of established pseudo-types for improved transduction efficiency [43].

**Table 1 jcm-13-07385-t001:** Overview of AAVs with tropisms for specific different cells of the retina. The AAV types that are reported to transduce retinal target cells either after direct subretinal injection (SRI), suprachoroidal injection (SCI), or intravitreal injection (IVI) into eyes are summarized in the table. An ‘x’ indicates that the respective cell type was reported to be transduced by the corresponding AAV type via the specified injection method.

AAV	Ganglion			Müller			Rod/Cone			RPE			References
	IVI	SRI	SCI	IVI	SRI	SCI	IVI	SRI	SCI	IVI	SRI	SCI	
2/2	x	x		x	x								[33,34,35,58,63,64,65,66]
2/6	x			x							x	x	[31,66,67]
2/8	x	x		x	x			x	x				[31,34,57,65,68,69,70,71]
2/9				x	x			x	x				[31,65,68]
9/PHP.eB	x			x				x					[34]
2/ShH10				x									[72]
2/7m8	x	x		x	x		x	x		x	x		[73,74,75,76]
2/Y-F mutants	x	x		x	x		x	x		x	x		[73,77,78,79]
2/NHP mutants	x			x			x			x			[61]
2/K9 mutants	x			x			x			x			[60]
2/GL	x	x		x	x		x	x		x	x		[74]
2/NN	x	x		x	x		x	x		x	x		[74]
DJ	x			x			x						[80]
2/1								x	x		x	x	[31,64,81]
2/4								x			x		[19,81,82]
2/5								x					[19,63,64,65,68,69,83]
2/7								x					[65,68]
8								x	x		x	x	[35,76,84,85]
v128		x			x			x	x		x	x	[76]

Thereby, two methods were primarily used for the retina: rational design by induction of mutations addressing the amino acids, specifically tyrosine and threonine residues, in AAV capsid proteins followed by subsequent screening for enhanced infectivity [77,78]; directed evolution (DE), a technique that mimics natural evolution to engineer capsid proteins with improved or desired traits by subjecting them to iterative cycles of mutation and selection [34,60,61,72,74,75,80]. Concrete examples for engineered AAV mutants will be discussed later in the text (see Section 3).

## 3. Obstacles in AAV-Based Retinal Gene Therapy

Despite the success that AAVs are experiencing in retinal gene therapy, many challenges and questions remain open. Though tropism is central to the usage of AAV in retinal gene therapy, the understanding of how this viral family enters cells selectively to deliver its genetic load remains elusive [86,87]. As previously stated, cell and tissue tropism are defined by the capsid make-up, which for AAVs composes proteins VP1, VP2, and VP3 [87]. However, how capsid proteins interact with surface factors leading to varying host cell specificities as well as how capsid exchanges and modifications can confer higher efficiencies remains largely unelucidated [86,87,88,89].

According to the current understanding, AAV capsids can initiate primal attachment through an array of charged glycans, which are serotype-specific [89,90,91,92,93,94]. In 2016, the AAV receptor (AAVR) was identified and shown to be pan-serotypically essential for the internalization and subsequent transport of AAVs to the Golgi [95,96]. Subsequently, an AAV enters the nucleus through endocytic pathways. Building up on this discovery, GPR108, a G protein-coupled receptor located in the Golgi apparatus, was recently identified as a crucial entry factor for AAV transduction [87]. However, the specific post entry steps involving GPR108 are not well characterized. Notably, there are exceptions observed in certain AAV serotypes. AAV5 is thought to be GPR108-independent but necessitates AAVR for infection, while AAV4 and related serotypes are AAVR-independent but require GPR108 [86,87,97]. Adding to this, the endoprotease furin was discovered to be a restrictive host factor for AAV4 entry in 2023 [89]. Nonetheless, despite considerable divergencies, numerous studies highlight the Golgi apparatus as a critical facilitator of AAV nuclear import [89,98,99,100].

Although this research is in its infancy, it is of great interest since it may contribute to further restrict viral selectivity and minimize off-target transduction, which is a common main endeavor in AAV-based gene therapy [86].

On a higher level, another major obstacle in AAV-based retinal gene therapy is the transport of the viral vectors to deep layers of the retina, such as the PL or RPE. There are four general ocular delivery routes: subretinal injection (SRI), suprachoroidal injection (SCI), intravitreal injection (IVI), and topical administration (Figure 2).

To deliver genetic information to the cells of PL or RPE, SRI is commonly the most used methodology [101] (Table 1). That is because SRI has shown leading efficiency in transferring AAVs for strong transgene expression in all retinal layers, and particularly in the PL or RPE layers [27,102]. In this procedure, the viral particles are injected with a syringe into the space between the photoreceptors and the RPE (Figure 2). To gain access to this area, retinotomy is required, which describes a controlled cut-through across ILM and retina tissue with the syringe [103]. Additionally, the injected viral liquid creates a localized retinal detachment or “bleb”, from which the viral particles diffuse through the retinal tissue [102,104]. Since the eye is a relatively immune-privileged organ, as initially described, viruses are typically not exposed to host antibodies, which allows multiple administrations without considerable immune responses [105]. However, there are many problems that were observed in the practical use of SRI, such as risks for permanent retinal detachment and progression of cataracts [27]. Furthermore, subretinal delivery induces a temporary detachment of the retina, which can result in harm to the photoreceptors. Particularly, detachment of the fovea—an indentation in the retina that comprises the highest rod and cone density and therefore shows strongest visual acuity—can lead to persistent decline in visual capacity or even the development of a hole in its surrounding retinal area, named macula [104]. Additionally, the movement of subretinal fluid within the bleb may be unpredictable and the size and placement of the bleb can affect the therapeutic outcome. Since transduction only occurs in tissue exposed to the virus-containing fluid, multiple retinotomies may be necessary to target the desired area, which increases the risk for afore described impairments [101]. Moreover, the backflow of cellular material into the vitreous through the retinotomy can diminish efficiency of transduction and has been recorded to potentially induce the formation of secondary membranes in the epiretinal space [106].

In the search for less disruptive alternatives, the novel method of suprachoroidal injection was also explored to access PL or RPE layers. SCI involves the injection of a liquid into the suprachoroidal space, a recently discovered space between choroid and the inner side of the sclera, that can be expanded by injection of volume across the sclera (Figure 2) [107]. The injection can be performed with standard needles but demands a highly skilled operator, which is why special transscleral microneedles or microcannulation with a flexible microcatheter are being commonly used [107,108]. The only SCI method with FDA clearance is a sclerotomy, a cut through the sclera, with subsequent microcannulation mediated injection into the suprachoroidal space. Because standardized easy-to-handle methods are not available, SCI is still not of common use in clinical practice. SCI has shown potential to be utilized for AAV-based retinal gene therapy, since different studies presented efficient transfection of RPE or PL cells using AAV vectors [31,35,84,85] (Table 1). However, a recent study among them elucidated transfection efficiency differences of the serotype AAV8 with SCI in comparison to SRI in non-human primate retina [84]. Although SCI resulted in peripheral transduction of the RPE one month after infection, it was more diffuse than with SRI and did not involve photoreceptor cells. Additionally, inflammatory cell accumulation in the retina and choroid were more pronounced, and transgene expression diminished after two to three months without SRI. It was reasoned that shortcomings in SCI facilitated retinal transduction may originate from rapid blood flow and clearance of viral particles from choroid capillaries adjacent to the Bruch’s membrane [84,101]. Furthermore, even though SCI circumvents retinotomy and therefore is less invasive than SRI, it is still a surgical intervention that may cause suprachoroidal hemorrhage and retinal perforation [101]. A non-surgical alternative would be to topically rinse liquids onto the cornea (Figure 2). However, it cannot be utilized in retinal gene therapy, since AAVs cannot reach the retina when topically applied [109].

Lastly, intravitreal injection—in comparison to SRI and SCI—is the least invasive surgical AAV administration route and has been employed to transduce cells of the inner retinal layers, such as Ganglion or Müller cells, with common AAV serotypes (Table 1) [27]. Due to its simplicity, IVI procedures are a standard practice in ophthalmology. IVI can be performed in the office by setting the patient under local topical anesthesia and subsequently injecting the solution into the vitreous [101] (Figure 2). Since thereby the retina is not touched, it incurs little procedure-related dangers. The proven risks are the disruption of viral vector integrity by neutralizing antibodies (NABs) in the vitreous space through humoral immune responses and volume losses due to dilution of vectors into the vitreous fluid after injection [72,110,111,112]. Next to its generally advantageous safety profile, IVI also bears the potential to achieve larger retinal surface coverage [113].

Despite this, the arguably most challenging disadvantage of IVI remains that injection of AAVs into the vitreous results repeatedly in transductions limited to the inner retina whereby the inner limiting membrane (ILM) was identified to be the diffusive barrier hindering AAVs from further penetration [41,84,113,114]. Therefore, AAV serotypes are most often delivered to the outer retina with invasive methods, such as SRI or SCI (Table 1). Nonetheless, different strategies were pursued to overcome this obstacle with the aim of enabling IVI delivered AAVs to reach deeper layers of the retina, such as PL or RPE.

One approach is the mild enzymatic digestion of the ILM barrier to generate pores for enhanced AAV penetration. For different enzymes, particularly proteases and glycosidases, enhanced IVI-based transgene expression in cells of the RPE and PL along with reduced immune responses were demonstrated for various AAV serotypes [41,115]. Notably, such results were only presented in rodent eye models, which physiologically differ significantly from human eyes and higher animal eyes, such as porcine or primate eyes (Table 2). Hence, transferring the method to them may pose challenges due to e.g., variations in ILM thickness, vitreous chamber size and retinal characteristics [58,113,116]. Additionally, other studies investigating the use of enzymes in the vitreous have reported rare ocular adverse events, such as inflammation, reduced visual acuity, and other visual abnormalities [117]. These findings suggest caution against the clinical use of intravitreally administered enzymes [113,117]. To circumvent ILM disruption, exosome-associated AAVs (exo-AAVs) were also tested. Exo-AAVs were shown to transduce more effectively retinal cells in contrast to non-exosome-complexed AAVs, but only did so primarily in the inner retinal layers and to lesser extent in outer layers [118,119]. Additionally, these studies were conducted solely in rodent eyes and the exact mechanism of exosome mobility and its contribution to increased cellular tropism is not yet known [113].

Historically the most intensively studied method, which also avoids the need for ILM destruction, is the technology of capsid engineering for improved retinal AAV diffusivity. The first examples for rational design were individual and combinatory tyrosine (Y) to phenylalanine (F) substitutions in AAV2/8 and AAV2/2 capsids generated in 2009 and 2011 [77,78]. Later, different parental capsids, such as AAV2/5 and AAV2/9, were also used to create established Y-F substitutions as well as to introduce additional threonine (T) to valine (V) substitutions on established Y-F mutants [120,121]. Generally, Y-F mutants were confirmed to efficiently transduce multiple cell layers, including deeper ones, such as PL or RPE, after intravitreal administration in murine [73,77,78] and canine [79] eyes (Table 1). Notably, when given to isolated explants of higher organisms, such as macaque or human retina, the transduction efficiency of the Y-F mutants was significantly reduced [73]. Another study recently presented a rationally designed version of AAV8, namely AAVv128, which was shown to infect the RPE layer in canine eyes after IVI [76]. In vivo DE, on the other hand, was first employed on the challenge of ILM passaging in 2013 [75]. Previously, in 2009, DE-designed mutants for neuronal tissues were screened for their utility in IVI-mediated retinal transduction, whereby AAV2/ShH10 appeared to selectively transduce Müller cells, but not cell layers below, in rat eyes (Table 1) [72]. Similarly, in later years, already-available engineered capsid variants—such as AAV9/PHP.eB or AAV-DJ—were tested for their utility in ILM penetration but with IVI mediated transduction either limited to Ganglion and Müller cells (AAV9/PHP.eB) or not further than photoreceptor cells (AAV-DJ) (Table 1). Moreover, these studies only evaluated the vectors in mouse eyes, which differ substantially from higher eyes, such as human, primate, or porcine eyes [34,80] (Table 2).

Nevertheless, in 2013, the first retinal in vivo DE setup was presented, selecting AAV mutants for enhanced diffusivity across the ILM barrier in mice after IVI. Thereby, an AAV2 variant, namely AAV2/7m8, which could transfect the entire retina in murine eyes but failed to reach the RPE layer in primate eyes after IVI was generated [75] (Table 1). Following evaluations of the vector performance additionally confirmed this finding and demonstrated that AAV2/7m8 transfects ganglion and RPE layers when intravitreally injected into canine eyes [74,76]. It was also presented that AAV2/7m8 obtains tropisms for all retinal cell types and can diffuse and confer genetic cargo even to cells of the RPE in primate or human retina if administered directly onto retina explants [43,73]. Building up on this research, similar DE-based approaches were explored in 2021 using murine eyes [74] and other eye models, such as canine [60] or primate [61] eyes. AAV2/GL and AAV2/NN were DE-engineered by repetitive screening rounds in murine eyes. They showed widespread retinal transduction of all retinal layers, even layers of the outer retina, such as PL or RPE, across different eye species—such as mouse, dog, and non-human primate eyes—after IVI [74] (Table 1). Likewise, AAV2/K9 and AAV2/NHP variants were DE-generated using canine eyes and primate eyes, respectively. Both showed efficient transduction of primate retina, especially in cells of the PL and RPE, after IVI [60,61] (Table 1). Furthermore, AAV2/NN and AAV2/GL as well as AAV2/K9 and AAV2/NHP variants were proven to have tropism for various retinal cell types in ex vivo human retina explants [43,74].

Although most of them have preferences in terms of transduction strength for some retinal cell types, none of the capsid engineered mutants exhibit sole specificity to one certain cell type without any weak to moderate tropism for another (Table 1). The only measure under investigation to solve this issue and further restrict next-generation mutants’ tropism is an additional engineering-like promoter setting transgene expression under the control of a human rhodopsin promoter in AAV2/NN and AAV2/GL variants for photoreceptor-specific transduction [62,74]. Additionally, on a general note, even if some clinical trials are currently ongoing using next-generation variants (NCT03316560, NCT02416622, and NCT03748784), clinically more utilized standard AAV serotypes and pseudotypes, like 2/1 or 8, still rely substantially on invasive injection methodologies—such as SRI or SCI—to address deeper retinal layers (Table 1). Hence, diffusion of them across the ILM to the outer retina after IVI remains a major challenge, especially in higher eye models, such as human or porcine eyes, with increased cell density and larger sized vitreous (Table 2).

**Table 2 jcm-13-07385-t002:** Anatomic differences between the ocular systems of species. Quantitative information was obtained/adapted from Mulari et al. and Schnichels et al. [116,122]. To the best of our knowledge, the RPE thickness of porcine eyes has not been measured so far, as indicated by not available as ‘N/A’.

	Mouse	Monkey	Pig	Human
Size	3 mm	20 mm	24 mm	24 mm
Retinal thickness	204 μm	292 μm	300 μm	310 μm
Ganglion cell density	4000 cells/mm^2^	3100 cells/mm^2^	6000 cells/mm^2^	5700 cells/mm^2^
Peak cone density	1.8 × 10^4^ cells/mm^2^	14 × 10^4^ cells/mm^2^	3.9 × 10^4^ cells/mm^2^	15 × 10^4^ cells/mm^2^
RPE thickness	18 μm	16.5 μm	N/A	1–2 μm

Therefore, a generic transportation platform that allows non-invasive IVI-based delivery of AAVs to the outer retinal layers in complex eyes and likewise enables controlled transduction of AAV cargos to specific target cell populations would be of significant relevance to the field.

Exosomes have shown a strong capacity to transport various AAV types across biological barriers, such as the blood-brain barrier, effectively targeting underlying neuronal tissue [123]. However, as previously discussed, exosome-mediated IVI delivery of AAVs has so far been studied only in rodent eyes, which differ significantly from human eyes [118,119]. While exosomes show considerable promise, further research is needed to adapt them to the stated requirements of a retinal delivery platform.

In contrast, most nanoparticle technologies, despite their demonstrated compatibility with diverse biological cargo—like small drugs or nucleic acids—and their efficacy in ocular delivery have largely been unexplored for viral delivery [124]. An exception is magnetic nanoparticles (MNPs), which have recently garnered interest due to their remote controllability via magnetic fields and their proven versatile compatibility with various AAV types [125,126]. The potential of MNPs to serve as the universal targeted retinal AAV delivery platform in question will be examined in the following section.

## 4. Magnetic Nanoparticles for AAV Delivery

Magnetic nanoparticles were most intensively studied as delivery vehicles for viruses in vitro, i.e., on cultures of neuronal cells, and to a lesser extent in ex or in vivo organs and tissues [127,128]. The methodology of MNP-mediated virus delivery commonly builds on the initial MNP–viral vector complexation by electrostatic interactions or functionalized binding agents as well as subsequent concentration of such complexes onto cells, tissues, or organs under an external magnetic field formed by permanent magnets [129] (Figure 3). Usually, the MNPs contain a magnetic solid core composed of, e.g., iron oxide, and a coating of a charged modifying agent, like cationic polyethylenimine (PEI), to adjust surface polarity for optimized complexation. Additionally, functionalization of MNPs and/or viruses for ligand–ligand interactions, e.g., streptavidin–biotin binding, were utilized [130,131,132]. Nevertheless, such assemblies necessitate extensive chemical alterations of the viral vectors and/or the nanoparticles, thereby restricting their practicality. Conversely, the non-specific pairing of viral vectors and MNPs through electrostatic interactions presents a versatile approach, as it eliminates the need for additional manipulation of the particles and/or virus [129].

In vivo experiments with rodent models have shown that MNP-based virus delivery can enhance transduction efficiency within complex organs, such as the brain [125,129,130] and intestine [133] (Table 3). Additionally, it was affirmed that they reduced the total required viral dose for transduction and succeeded in traversing multiple biological diffusive barriers [127,129].

In the context of eyes, other bioactive compounds, like drugs, nucleic acids, or stem cells, were tested for in vivo magnetic retinal delivery within rodent eyes [134,135,136,137], but viruses were not. To the best of our knowledge, MNP-based viral retinal delivery has been tested exclusively in ex vivo settings but not using in vivo models [125,126] (Table 3).

**Table 3 jcm-13-07385-t003:** Selected reports on virus delivery with magnetic nanoparticles (MNPs) in vivo and ex vivo. SPION: superparamagnetic iron oxide nanoparticles; PEI: polyethylenimine; IOP: iron oxide particles; AEEA: N-(3-trimethoxysilylpropyl)diethylenetriamine; HSPG: heparan sulfate proteoglycan.

Species	Tissue or Organ	MNPs	Modifying Agent	Virus Type	Reference
Mouse	Brain (in vivo)	SPION	PEI, Streptavidin	Biotinylated adenovirus	[130]
Mouse	Tumor xenografts (in vivo)	IOP	PEI	Adenovirus	[138]
Mouse	Blood vessels (in vivo, ex vivo)	CombiMag, TransMag (Chemicell, Berlin, Germany)	-	Lentivirus	[139]
Mouse	Intestine (in vivo)	IOP	N-hexanoyl chitosan	Adenovirus	[140]
Rat	Intestine (in vivo), rlood vessels (in vivo)	SPION	PEI	Retrovirus, Adenovirus	[133]
Rat	Brain (in vivo)	AdenoMag (OZ Biosciences, Marseille, France)	-	Adenovirus	[129]
Mouse	Brain (in vivo), retina (ex vivo)	SuperMag Silica beads (Alpha Biobeads, San Diego, CA, USA)	AEEA	Rabiesvirus, Lentivirus, AAV	[125]
		ViroMag (OZ Biosciences, Marseille, France)	-		
Mouse	Lung (in vivo)	Magnetite particle	HSPG	AAV	[131]
Pig	Retina (ex vivo), eye (ex vivo)	FluoMag-V (OZ Biosciences, Marseille, France)	-	AAV	[126]

In 2018, the MNP-based delivery of AAVs to the inner plexiform layer of mouse retina explants was presented [125]. While this study demonstrated the applicability of the MNP methodology to retinal tissue, it did not explore its potential to propel AAVs across the ILM barrier in clinically relevant, larger eye models. A recent study addressed this issue by successfully using MNPs in ex vivo porcine eyes to magnetically transport various AAV types across the ILM for selective transduction of target retinal layers, like the photoreceptor and RPE layer after IVI [126]. Notably, AAVs efficiently transduced retinal layers against their known layer-specific tropic preferences, which the authors hypothesized to be due to an MNP-mediated enhanced local viral load and residence time that may enforce on-target transduction [126]. Putative vitreous NAB responses to IVI-administered AAV-MNP complexes were not investigated in the study [126].

Furthermore, neither that study nor another one characterizing IVI-injected, cargo-free MNPs in human-mimetic porcine eyes reported on MNP clearance from the eye with prolonged magnetic actuation [126,141]. However, it is important to note that a magnetic exposure period exceeding six hours post-IVI was thereby not evaluated. Additionally, it was congruently shown that MNPs with or without viral cargo fail to pass the cornea in pig eyes under a 24 h external magnetic field when topically administered [126,142].

Besides this, MNPs are commonly used as contrast agent for magnetic resonance imaging (MRI) in clinics [143,144]. For ocular applications, it was elucidated that MNPs can be detected by MRI scanners in vivo and are cleared from the ocular system within weeks after IVI in rodent eyes [145]. It was also presented that IVI of MNPs does not result in toxic effects on retinal structure, photoreceptor function, aqueous drainage, or intraocular pressure [143,145].

In summary, the findings indicate that MNPs can effectively transport various AAVs across the ILM to deeper retinal layers following IVI in higher eye models and suggest a potential MNP-mediated enhancement in AAV transduction efficiency. They also emphasized that MNPs are safe to use and trackable in vivo with standard clinical devices.

To this end, it is important to recognize that using also MNPs has limitations, such as their restricted ability to cover larger three-dimensional areas, as they tend to be concentrated in a two-dimensional plane under an external magnetic field. Additionally, retaining the particles within the target organ becomes challenging once the magnetic field is removed [143,146,147]. Moreover, MNPs commonly tend to form aggregates in formulations [147,148]. Lastly, the therapeutic efficacy of MNP-based treatments is constrained by the limited intervals, intensities, and exposure times of magnetic fields that patients can safely tolerate [143,149].

For MNP-based retinal AAV delivery, the tendency of MNPs to align in a two-dimensional diffusive front and to be cleared from organs post-magnetic actuation could be considered advantageous as it supports the precise and minimally invasive targeting of specific retinal layers on a microscale. Nevertheless, MNP aggregates were consistently observed to form even within retinal tissues after injection. So far, only precautious measures, like ultrasonication of MNP dilutions, were conducted aiming to minimize aggregation before injection [125,126]. Thus, preventing aggregation throughout the entire MNP treatment remains a central obstacle. Nonetheless, a recent study demonstrated that MNPs within a nanoparticle-laden droplet suspended in a viscous polymer medium could be precisely extracted and manipulated using an external magnetic field [150]. This approach minimizes aggregation and allows for controlled MNP diffusion, offering a potential solution to improve the targeting and remote controllability of MNPs in retinal AAV delivery.

## 5. Conclusions and Future Perspectives

Current studies on MNP-based viral delivery to ocular systems, particularly using in vivo models, remain elusive. Investigations employing eye models with anatomical similarities to the human eye, such as those from pigs or non-human primates, could provide more translational insights into the potential of MNP-based retinal gene therapies. Furthermore, the application of MRI in MNP-guided viral delivery warrants investigation given its clinical relevance and demonstrated effectiveness in tracking cargo-free MNPs following intravitreal injection. Additionally, controlled removal of MNPs after treatment is of interest. Therefore, exploring the feasibility of guiding MNPs from the retina to the choroid and blood system with prolonged magnetic exposure could be valuable. Considering safety, these studies should be conducted within the tolerance limits of biological systems to prolonged MRI or magnetic field exposure—another challenge that future research must address. Moreover, the immune response to injected viruses when complexed with MNPs requires additional assessment for the approach to be putatively translated into clinical practice. Ultimately, as it is substantial to all MNP technologies, efforts must be made to develop methodologies that minimize MNP aggregation.

## Figures and Tables

**Figure 1 jcm-13-07385-f001:**
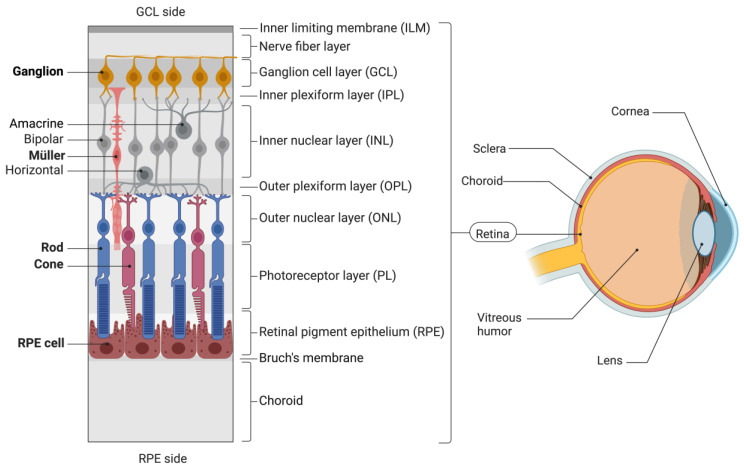
Organization of ocular systems and cell-type-specific retinal tropism of AAVs. The retina is depicted upwards, with the inner retina (GCL side) at the top and the outer retina (RPE side) at the bottom. Retinal cells with reported AAV tropisms (Table 1) are highlighted and depicted in bold. Schematics were created using Biorender.com.

**Figure 2 jcm-13-07385-f002:**
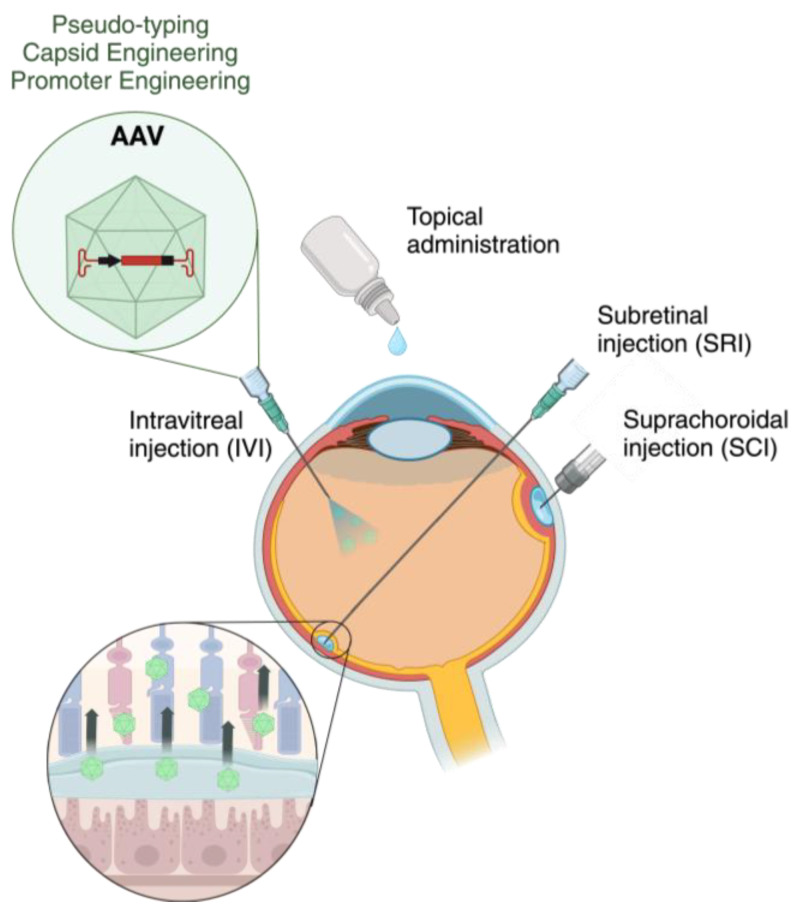
Common injection and vector engineering methods in AAV-based retinal gene therapy. While subretinal injection (SRI) and suprachoroidal injection (SCI) rely on the diffusion of AAVs from the injected bleb at the retinal pigment epithelium (RPE) side toward the ganglion cell layer (GCL), intravitreal injection (IVI) depends on the diffusion of AAVs across the vitreous and inner limiting membrane (ILM) from the GCL toward the RPE side. Schematics were created using Biorender.com.

**Figure 3 jcm-13-07385-f003:**
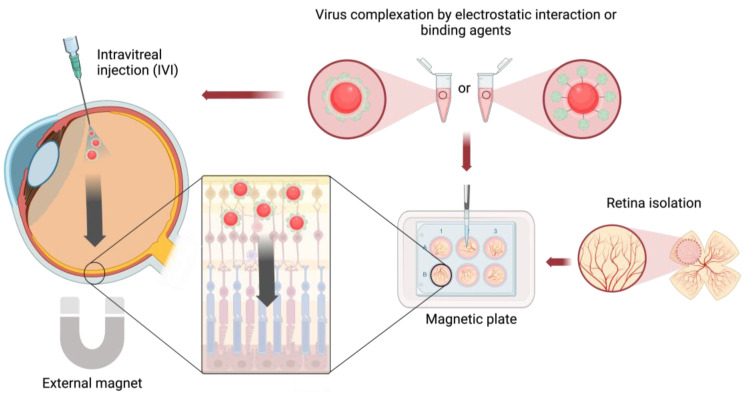
Principles of MNP-based virus delivery at the example of retina explants and intravitreal eye injections. Complexation of MNPs and viruses is executed by electrostatic interaction or binding agents. Such complexes are then applied via IVI to eyes (**left**) or pipetted onto retinal explants (**right**) upon permanent magnetic field induction for remotely controllable viral delivery. Schematics were created using Biorender.com.

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
