# Peer review of "Challenges in AAV-Based Retinal Gene Therapies and the Role of Magnetic Nanoparticle Platforms"

_jcm, 2024, doi:10.3390/jcm13237385_

Round 1
Reviewer 1 Report
Comments and Suggestions for Authors
The current manuscript aims to describe the challenges in AAV-based retinal gene therapies and the role of magnetic nanoparticle platforms. Although the topic is interesting in its scientific field, there are some issues that require the authors’ attention to improve the quality of this particular manuscript before further consideration for publication in a high-quality journal “JCM”.
Specific comments:
1. Both Figure 1 and Figure 2 present general ocular images. Please provide specific examples or conceptual graphs of AAV-based retinal treatment.
2. Table 1 looks messy. It is recommended that the authors should reorganize the classification.
3. This article emphasizes the use of AAV for retinal gene delivery. Please explain the rationale for this selection. Whether any specific advantages of AAV are beneficial for retinal applications or existing challenges are faced by other viral vectors that make them less suitable? Please clarify.
4. Why this study only focus on magnetic nanoparticles (MNPs) [rather than other nanoparticle platforms] for AAV delivery? Please justify.
5. Please have a comprehensive comparison of the therapeutic performance (such as transduction efficiency and retinal penetration) between MNP-AAV complexes and other delivery methods.
6. As mentioned in Section 2, the eye particularly is an attractive target due its immune-privileged nature. Nevertheless, this important scientific claim is not supported by any documentation. If possible, please consider the inclusion of the following relevant case study (DOI: 10.1002/jbm.a.31619) in the reference list to strengthen manuscript quality and attract more attention from broad readers.
Reviewer 2 Report
Comments and Suggestions for Authors The article meets the required criteria for this type.It is processed according to established criteria,
provides an overview of the latest scientific knowledge
in the field of AAV-Based Retinal Gene Therapies
and the Role of Magnetic Nanoparticle Platforms.
The article is prepared clearly and clearly for the reader.
No citations are published in pictures 1 - 3, please add or explain (authorship).
Comments on the Quality of English Language I have no comments.
Author Response
Reviewer 2: Comments and Suggestions for Authors
The article meets the required criteria for this type. It is processed according to established criteria, provides an overview of the latest scientific knowledge in the field of AAV-Based Retinal Gene Therapies and the Role of Magnetic Nanoparticle Platforms. The article is prepared clearly and clearly for the reader.
Authors: We thank the reviewer for the positive and constructive feedback. In the following, we describe how we addressed each point raised by the reviewer in our revised manuscript.
Specific comments:
Reviewer 2.1. No citations are published in pictures 1 - 3, please add or explain (authorship).
Authors: We thank the reviewer for pointing out this issue. We created all the schematic illustrations by using Biorender.com. To clarify this, we referenced the source in all figure captions accordingly.
In revised figure captions: “…Schematics were created using Biorender.com.”
Reviewer 3 Report
Comments and Suggestions for Authors
In this review, Siontas et al. talked about the status adeno associated viruses (AAVs) vector system for the treatment of retinal disorder. While AAV offer promising therapeutic potential, the clinical application is limited by challenges, such as invasive administration and tropism. The author proposed a new strategies, involving the combination of magnetic nanoparticles (MNPs) with AAVs for improved viral delivery and therapeutic efficacy. Overall, this is an interesting review. However, there are a several technical issues that need to be addressed (see comments below).
1. In Page 2 Line 82, the sentence “The retina is composed of multiple layers of neural tissues…” is not appropriate to be in the last sentence of the paragraph. This paragraph focuses on retina features for gene therapy and the sentence is not relevant. Please consider relocating to a more appropriate position.
2. The manuscript uses “i.e.” improperly. Please check and remove “i.e.” where it is unnecessary. For examples, in the following sentence “Thanks to the technique of pseudo-typing, variants were generated that i.e., more preferably transduce cells of the PL and RPE”. Here, “i.e.” should be omitted.
3. What is the mechanism for MNP clearance after administering in the eye?
4. In Page 12 Line 426, the author mentioned “frequency”. How does the magnetic filed frequency affect MNP based treatments?
5. There are many types of MNP and various techniques to fabricate them. What specific type of MNPs were used in combination with AAV? Additionally, MNP aggregation is a significant concern. What is the potential methodology to minimize MNP aggregation.
Author Response
Point-by-Point response to the comments of the Reviewer #3
Reviewer 3: Comments and Suggestions for Authors
In this review, Siontas et al. talked about the status adeno associated viruses (AAVs) vector system for the treatment of retinal disorder. While AAV offer promising therapeutic potential, the clinical application is limited by challenges, such as invasive administration and tropism. The author proposed a new strategies, involving the combination of magnetic nanoparticles (MNPs) with AAVs for improved viral delivery and therapeutic efficacy. Overall, this is an interesting review. However, there are a several technical issues that need to be addressed (see comments below).
Authors: We thank the reviewer for the positive and constructive feedback. In the following, we describe how we addressed each point raised by the reviewer in our revised manuscript.
Reviewer 3.1. In Page 2 Line 82, the sentence “The retina is composed of multiple layers of neural tissues…” is not appropriate to be in the last sentence of the paragraph. This paragraph focuses on retina features for gene therapy and the sentence is not relevant. Please consider relocating to a more appropriate position.
Authors: To address this concern. we moved the sentence to a more relevant position in the section 2 as follow:
In page 2, line 93: “For instance, various neural cells in the multilayered retinal tissue – composed of the inner limiting membrane (ILM), ganglion cell layer (GCL), inner plexiform and nuclear layer (IPL and INL), as well as outer retina layers containing the outer nuclear layer (ONL), the photoreceptor layer (PL) and retinal pigment epithelium (RPE) – can be individually targeted only by the choice of respective viral vectors that obtain matching tropism specificities (Figure 1).”
Reviewer 3.2. The manuscript uses “i.e.” improperly. Please check and remove “i.e.” where it is unnecessary. For examples, in the following sentence “Thanks to the technique of pseudo-typing, variants were generated that i.e., more preferably transduce cells of the PL and RPE”. Here, “i.e.” should be omitted.
Authors: We thank the comment and omitted “i.e.” at unnecessary locations in the text.
Reviewer 3.3. What is the mechanism for MNP clearance after administering in the eye?
Authors: We appreciate the question, but must point out that the precise mechanism is, to the best of our knowledge, unknown yet. In a previous paper (Raju et al (2012) Clinical Exper Ophthalmology, doi:10.1111/j.1442-9071.2011.02651.x.), it was shown that MNPs can be detected by MRI scanners in rodents one hour and one day after intravitreal injection. When investigating the brain, liver, spleen or kidney at 5 weeks after intravitreal injection using ex vivo high-resolution MRI of the explanted organs, the authors reported that MNPs were not observable by MRI. They hypothesised that macrophage engulfment of the MNPs may have contributed to this observation, but an actual explanation remains elusive.
To reflect this comment, we added new sentences in the revised manuscript
In page 12, line 516, “Besides this, MNPs are commonly used as contrast agent for magnetic resonance imaging (MRI) in clinics [140,141]. For ocular applications, it was elucidated that MNPs can be detected by MRI scanners in vivo and are cleared from the ocular system within weeks after IVI in rodent eyes [142].
Reviewer 3.4. In Page 12 Line 426, the author mentioned “frequency”. How does the magnetic filed frequency affect MNP based treatments?
Authors: We apologise for the confusion. We intended to describe that how often the patient being put into a magnetic field, not the physical parameter of magnetic field frequency, could influence the translatability of MNP-based treatments into clinical use. Therefore, we changed the line in the section 4 as follows:
In page 12, line 532, “Lastly, the therapeutic efficacy of MNP-based treatments is constrained by the limited intervals, intensities, and exposure times of magnetic fields that patients can safely tolerate [140,146].”
Reviewer 3.5. There are many types of MNP and various techniques to fabricate them. What specific type of MNPs were used in combination with AAV? Additionally, MNP aggregation is a significant concern. What is the potential methodology to minimize MNP aggregation.
Authors: We would like to point out that all MNP types for AAV delivery are listed in Table 3. Moreover, we have added a very recent proposed methodology (new reference #150) to minimise MNP aggregation in section 4 which may contribute to the field of MNP-based retinal AAV delivery. We have additionally discussed that, to date, this obstacle has not been overcome and only thoroughly been tried to be addressed by precautious ultrasonication of MNPs before injection:
In page 12, line 538, “Nevertheless, MNP aggregates were consistently observed to form even within retinal tissues after injection. So far only precautious measures, like ultrasonication of MNP dilutions, were conducted aiming to minimize aggregation before injection [122,144]. Thus, preventing aggregation throughout the entire MNP treatment remains a central obstacle. Nonetheless, a recent study demonstrated that MNPs within a nanoparticle-laden droplet, suspended in a viscous polymer medium, could be precisely extracted and manipulated using an external magnetic field [150]. This approach minimizes aggregation and allows for controlled MNP diffusion, offering a potential solution to improve the targeting and remote controllability of MNPs in retinal AAV delivery.”
Round 2
Reviewer 1 Report
Comments and Suggestions for Authors
The revised version has adequately addressed most of the critiques raised by this reviewer and is now suitable for publication in "JCM".